# Study on the Microstructure of Polyether Ether Ketone Films Irradiated with 170 keV Protons by Grazing Incidence Small Angle X-ray Scattering (GISAXS) Technology

**DOI:** 10.3390/polym12112717

**Published:** 2020-11-17

**Authors:** Hongxia Li, Jianqun Yang, Feng Tian, Xingji Li, Shangli Dong

**Affiliations:** 1School of Materials Science and Engineering, Harbin Institute of Technology, Harbin 150001, China; koalalhx@163.com (H.L.); yangjianqun@hit.edu.cn (J.Y.); 2Shanghai Synchrotron Radiation Facility, Zhangjiang Lab, Shanghai Advanced Research Institute, Chinese Academy of Sciences, Shanghai 201204, China; tianfeng@zjlab.org.cn

**Keywords:** PEEK, SIRM, damage mechanisms, GISAXS, irradiation

## Abstract

Polyether ether ketone (PEEK) films irradiated with 170 keV protons were calculated by the stopping and ranges of ions in matter (SRIM) software. The results showed that the damage caused by 170 keV protons was only several microns of the PEEK surface, and the ionization absorbed dose and displacement absorbed dose were calculated. The surface morphology and roughness of PEEK after proton irradiation were studied by atomic force microscope (AFM). GISAXS was used to analyze the surface structural information of the pristine and irradiated PEEK. The experimental results showed that near the surface of the pristine and irradiated PEEK exists a peak, and the peak gradually disappeared with the increasing of the angles of incidence and the peak changed after irradiation, which implies the 170 keV protons have an effect on PEEK structure. The influences of PEEK irradiated with protons on the melting temperature and crystallization temperature was investigated by differential scanning calorimetry (DSC). The DSC results showed that the crystallinity of the polymer after irradiation decreased. The structure and content of free radicals of pristine and irradiated PEEK were studied by Fourier transform infrared spectroscopy (FTIR) and electron paramagnetic resonance (EPR). The stress and strain test results showed that the yield strength of the PEEK irradiated with 5 × 10^15^ p/cm^2^ and 1 × 10^16^ p/cm^2^ was higher than the pristine, but the elongation at break of the PEEK irradiated with 5 × 10^15^ p/cm^2^ and 1 × 10^16^ p/cm^2^ decreased obviously.

## 1. Introduction

Polyether ether ketone (PEEK) is widely used as an electrical insulation material in the aerospace field due to its excellent thermal and mechanical properties. The aromatic rings in the PEEK backbone are responsible for its strength, heat and radiation resistance [1,2,3]. The pendant ketone group increases the spacing between the molecules, and the ether bond makes the main chain flexible. The excellent mechanical stability and radiation resistance of PEEK has made it a choice material in a number of applications in the space environment [4]. In some areas of application, e.g., in the nuclear industry or space research, the radiation resistance of PEEK is of major importance [5]. The irradiation causes degradation of polymeric chains, breakage of chemical bonds, generation of free radicals and release of gas degradation products [6,7,8]. Subsequent chemical reactions of transient highly reactive species lead to excessive double bonds [9], low-mass stable degradation products, large cross-linked structures and oxidized structures [10]. So far, although the related research on the PEEK irradiation effect and its damage mechanism have been reported at home and abroad, it is still in the beginning exploration stage. Only a few papers mainly choose the gamma ray, heavy ion and electron irradiation sources to test and evaluate the chemical properties after irradiation, the mechanical behavior and the mechanism of the irradiated PEEK, of which the irradiated damage is only several microns and has not been studied [11,12,13]. The mechanical behavior of polymer insulating materials, especially a deep understanding of tensile deformation behavior, can help to evaluate the behavior of the material in orbit [14]. In recent years, synchrotron radiation X-ray scattering technology provides an effective method for detecting the microstructure changes of polymer materials [15,16]. The results can provide theoretical basis for the development of light, high performance and low-cost polymer materials, and the technology have important academic value. Our group has studied the structure evolution mechanism of the Low-density Polyethylene (LDPE) and PEEK polymer after electron irradiation by small angle X-ray scattering (SAXS) technology. There are a large number of low-energy charged particles in space, and these particles cause great damage to the structure and performance of the material. The damage caused by low-energy proton irradiation is only on the surface. How the surface structure and properties change after irradiation and the impact on the overall structure and properties has not been studied yet. The changes in the internal crystals of the material will significantly affect the changes in material properties. The effect of surface structure damage on the overall structure and performance degradation of the material is of great significance to the study of material degradation mechanisms.

GISAXS is a powerful tool for studying the surface of the films and interface structures [17,18,19]. Due to the small incidence angle of grazing incidence, we usually detect the area given by the slender coverage area of X-ray beam on the sample. The horizontal beam width is usually about 0.5 mm, and the coverage area extends the full length of the sample along the beam direction. The typical GISAXS sample size is from 10 mm to 30 mm, so we detect several mm^2^ surface macroscopic areas, and the structural period is from 1 nm to 100 nm. In addition, the dispersion signal is proportional to the volume square of the irradiated sample area, and the 100 nm film on the 20 mm base is 10^6^ μm^3^. In contrast, the area detected by a typical transmitted SAXS beam is about 1 mm × 1 mm, which is less than one-tenth of the amount of scattering, and therefore less than one-hundredth of the intensity of scattering [20]. In addition, the substrate causes attenuation. When the 0.5 mm silicon wafer is irradiated at the speed of 10 keV, the transmittance will be reduced to 3%. And we can’t get information about the height of the membrane. So the grazing incidence is used to analyze the microstructure damage. The structure evolution mechanisms of PEEK after low-energy proton irradiation were studied by the advanced GISAXS technology. The damage was only on the surface of the irradiated PEEK.

In this paper, the surface microstructure damage mechanisms of the PEEK irradiated with 170 keV proton were studied. The tensile tests and AFM were used to analyze the stress-strain and surface morphology of PEEK. Synchrotron radiation grazing incidence small angle X-ray scattering (GISAXS), FT-IR and DSC were applied to analyze the structure and crystallinity change of the PEEK after 170 keV proton irradiation. We used Stopping and Range of Ions in Matter (SRIM) to calculate the incident depth and ionization and displacement absorbed dose of the PEEK, combined with the advanced GISAXS technology and other experimental methods to reveal the influence of the material surface damage on the overall structure and performance degradation.

## 2. Experimental Section

### 2.1. Materials and Equipment

The density of PEEK material films was 1.3 g/cm^3^ and the thickness was 50 μm. They were manufactured by the British Weiges Co., Ltd. (London, United Kingdom), and their molecular formula is [C_9_O_3_H_12_]_n_. The molecular structure diagram is shown in Figure 1. The 170 keV proton irradiation was performed by the experimental device of low-energy charged particles irradiation at Harbin Institute of Technology (Harbin, China).

### 2.2. Experimental Parameter

The experimental proton energy was 170 keV, the sample chamber was vacuum 10^−8^ Pa, the irradiation area was 5 × 5 cm^2^, the flux was 1 × 10^15^ p/cm^2^·s, and the fluence values were 1 × 10^15^ p/cm^2^, 5 × 10^15^ p/cm^2^ and 1 × 10^16^ p/cm^2^. The sizes of the dumbbell-shaped tensile samples were 12, 4 and 0.5 mm, respectively.

### 2.3. Microstructural and Mechanical Property Analysis

The AFM images were obtained by atomic force microscope with a multimode scanning probe microscope (Dimension Fastscan) produced by Bruker, Berlin, Germany. The maximum scanning range was 90 μm, the test temperature was −35–250 °C, the elastic modulus range was 1 MPa~100 Gpa, the adhesion force range was 10 pN~10 μN, the surface potential was ±10 V, and the precision was 10 mV. Moving the sample across the x-y plane, a voltage was applied to move the piezoelectric driver along the z-axis to maintain the same detection force, resulting in a three-dimensional image of the height of the sample surface. To evaluate mechanical performances of pristine and irradiated PEEK, elongation at break and tensile strength were tested by the MTS 810 material analysis and testing system of MTS (Shimadzu, Japan). The samples used for the tensile test were the dumbbell-shaped sample with a draw rate of 2 μm/s at room temperature. The length, width and thickness of the dumbbell-shaped tensile test samples were 12, 4, and 0.5 mm, respectively. The Fourier transform infrared (FT-IR) absorption spectra were used to analyze the microstructure of the samples, and the spectra were obtained in wavenumber range from 700 to 4000 cm^−1^ at every 2 cm^−1^ using a Magna-IR 560 spectrometer produced by Thermo scientific, Waltham, MA, USA. The free radicals before and after irradiation were analyzed by electron paramagnetic resonance (EPR) spectrometer (A200) from Bruker company, Berlin, Germany, the magnetic field ranged from 0 to 7000 G at a microwave frequency of 100 kHz. The melting and crystallization behavior were measured by differential scanning calorimetry (DSC) by a calorimeter (204 F1, Netzsch, Selb, Germany), the samples test temperature ranged from 30 to 400 °C at a heating and cooling rate of 10 °C/min. The grazing incident small angle X-ray scattering (GISAXS) measurements were performed on beamline BL16B1 of the Shanghai synchrotron radiation facility (SSRF) located in Shanghai Institute of Applied Physics, China. The Kohzu tilt stage was used as the GISAXS sample stage. The incidence angle of X-ray could be adjusted by the sample stage within an accuracy of 0.001°. In principle, working with an X-ray energy of 10 keV, the beam spot size at the samples was 0.5 mm × 0.5 mm. The sample detector distance was 5000 mm and the wavelength of incident X-ray was 0.124 nm. The schematic diagram of GISAXS test is shown in Figure 2. The GISAXS data was processed by FIT2D (London, UK).

## 3. Results

### 3.1. SIRM Calculation

The ionization and displacement effect are the two aspects of the interaction between high-energy particles and materials. The ionization effect causes the polymer structure to form an excited state and the chemical bonds are broken. The displacement effect causes C, H, O and other atoms to leave the original lattice position to migrate along the polymer chain or in the material, and the atoms will be captured and stabilized in the appropriate position. Understanding the sensitivity of polymer materials to the ionization/displacement effect is of great significance for predicting the performance of polymer materials in an irradiation environment.

The SRIM software simulates the evolution of ionization and displacement dose produced by the PEEK irradiated with 170 keV proton and the content of free radicals changes with the change of ionization and displacement dose, as shown in Figure 3. If it is assumed that the excitation process of electrons and atoms is irrelevant, the energy loss of incident particles can be attributed to the sum of ionization energy loss and displacement energy loss, and the expression is shown in Equation (1):(1)$$\frac{\mathrm{dE}}{\mathrm{dx}}=N\left[S_{n}\left(E\right)+S_{e}\left(E\right)\right]$$

Among them, *S_n_*(*E*) is the blocking cross section of nuclear blocking, *S_e_*(*E*) is the blocking cross section of electron blocking, and N is the atomic density of the material, so the particle range is shown in Equation (2):(2)$$R=\int_{0}^{R}-\frac{dE}{dE/dx}=\frac{1}{N}\int_{0}^{E}\frac{dE}{S_{n}\left(E\right)+S_{e}\left(E\right)}$$

It can be seen that the total range of ions only depends on the electron blocking cross section and the nuclear blocking cross section, while the nuclear blocking and electron blocking cross sections depend on the interaction of particle collisions. The absorbed dose irradiated by charged particles refers to the radiant energy absorbed by a unit mass of material. The mathematical expression of the absorbed dose of a single-energy charged particle is shown in Equation (3):(3)$$D=\frac{1}{\rho }S\cdot \Phi$$

Dose is the absorbed dose, *ρ* is the density of the irradiated material, *S* is the stopping power, and Φ is the spectral distribution of particle fluence versus energy. Ionization damage is characterized by linear energy transfer (LET). The absorbed ionization dose can be obtained by multiplying LET by the fluence Φ, and the calculation formula is as follows (4):(4)$$D_{i}\left(\mathrm{rad}\right)=1.6\times 10^{-8}LET\times \Phi$$

1.6 × 10^−8^ is the unit conversion factor.

Similar to the calculation of ionization damage, displacement damage is characterized by nonionizing energy loss (NIEL), and the calculation formula is as follows (5):(5)$$D_{d}\left(\mathrm{rad}\right)=1.6\times 10^{-8}NIEL\times \Phi$$

According to the above formula, the results of ionization and displacement absorbed dose can be calculated and combined with the results of the EPR test, the result of Figure 3d can be obtained. The content of free radicals increases with the increasing of ionization and displacement absorbed dose.

### 3.2. Surface Morphology and Roughness Analysis

The surface roughness values and 3D morphology images of pristine and irradiated PEEK with different fluences analyzed by AFM are presented Figure 4. The Figure 4d shows the roughness variation curves of pristine and irradiated PEEK in real-time. The roughness analysis of these samples was carried out for the chosen scan areas of 20 μm × 20 μm. Mean roughness (Ra) and root mean square (Rq) were calculated and are presented in Table 1. The PEEK film before irradiation (Figure 4a) exhibited an almost flat surface with some small bumps on it and its roughness parameters Ra and Rq were around 18.8 nm and 23.4 nm, respectively. When the PEEK was irradiated at 5 × 10^15^ p/cm^2^, the films (Figure 4b) displayed a moderately creased surface which is revealed by slightly higher roughness values. In this case, Ra as well as Rq were relatively higher, and equal to 20.8 nm and 26.1 nm, respectively. When the irradiation fluences further increased to 1 × 10^16^ p/cm^2^, the surface roughness values (Figure 4c) decreased slightly, the Ra and Rq decreased to 17.5 nm and 22.3 nm, respectively. The reason may be that when the irradiation fluence was 5 × 10^15^ p/cm^2^, the radiation damage on the surface of the PEEK material caused the roughness to increase, and when the irradiation fluence was 1 × 10^16^ p/cm^2^, the irradiation slightly etched the surface of the material to reduce the surface roughness. However, overall, as can be seen from Table 1, the surface roughness of PEEK before and after irradiation changed slightly. Generally speaking, the roughness of the material affected the GISAXS results which ignore the effect of the roughness.

### 3.3. Micrstructure Analysis Based on GISAXS

The GISAXS scattering patterns and intensity distribution curves with different incident angles for pristine and irradiated PEEK with different fluences are shown in Figure 5. The GISAXS technology provides a powerful tool for studying ordered interface and surface of materials [21,22]. The effect of roughness on the experimental results is very complicated. In this experiment, the AFM experiment was used to prove that the roughness after irradiation was affected slightly, so the effect of roughness on the GISAXS results was ignored. If the layered structure is parallel to the substrate, the dispersion peaks will be obtained along the surface normal in the plane of incidence. For a freely oriented layer materials, we get a ring diffraction peak. Because the scattered X-rays are blocked by the base, the ring peak only can be seen if the angle of departure is greater than zero. If the layered structure is partially oriented, the ring peak will become an arc peak. For the layered structure, we will observe the Bragg reflection in the direction parallel to the substrate surface. The layered structures with substrates and polymer membranes may be observed in disordered systems as rings or arcs. It can be seen from the Figure 5a, there appeared obvious scattering peak and it shows different shapes with different angles of incidence. The horizontal stripe diffraction pattern represents the crystal in the horizontal direction. This peak should be a layered crystal structure. The Figure 5b shows that there are multiple peaks, which may be multiple crystal structures. However, the GISAXS images of the PEEK irradiated by 170 keV protons with different fluences changed significantly, as shown Figure 5c–f. When the angles of incidence increases to a certain degree, the peak gradually changes and eventually disappears. This results combined with the SRIM calculation results, which show that irradiation makes the PEEK materials produce ionization effect and displacement effect, resulting in a large number of vacancies, which affects the crystal structure of the material. It is known that the X-ray incident on crystal materials will cause the Bragg diffraction of the crystals as given by the following formula:(6)2dsinθ=nλ
*n* is the diffraction order, *d* is the crystal spacing, *θ* is the diffraction angle and *λ* is the incident X-ray wavelength.

Crystalline phases can have their own formula to represent the long period, while giving a larger value L. The interphase distribution of the crystalline and amorphous regions constitutes this structure distance between two adjacent crystal regions. Therefore, when the incident wavelength is constant, the Bragg diffraction can occur at a smaller angle, which corresponds to a smaller scattering vector q. The Bragg diffraction peak generated by the long-period structure in a small angle range can represent the SAXS scattering peak of the pristine PEEK. According to the definition and the formula (6) of the scattering vector, the relationship between the long period and scattering vector can be obtained as follows:(7)q=2nπL
*q* is the scattering vector, *n* is the number of diffraction orders, and L is the long period of the crystalline polymer. According to formula (7) and the scattering peak position of the undisturbed PEEK, the long period of the pristine PEEK without stretching was 16.44 nm. The changes in long period of the pristine and the irradiated PEEK with the incident Angle are shown in Table 2. From the table, it can be seen that with the increasing of the incident angle, the long period of the PEEK decreased. Irradiation reduced the long period of the material and changes the internal crystal structure of the material.

### 3.4. EPR Analysis

The EPR spectra of pristine and PEEK irradiated with different fluences and the content of free radicals changed with fluences are shown in Figure 6a,b. In polymer materials, the displacement effect will cause some atoms in the molecule chain to change from the original position to bond breakage, resulting in the degradation of the molecular chain. Covalent bond breakage leaves the atoms in an unpaired electronic state and then generates free radicals [23]. The charged particles will interact with atoms in the outer layer of polymer, generating electron excitation and causing ionization, which may also generate free radicals in the molecular chain. It can be seen from Figure 6 that the PEEK after 170 keV proton irradiation produced a large amount of free radicals, and the content of the free radicals increased with the increasing of the fluences. The data were calculated by the Formula (8):(8)N=∬Sdsm
where N is the content of free radicals, *S* is the measured EPR spectrum, m is the quality of test sample. The g value was 2.0025 in this data, and the g value was stable and it did not change with the change of the irradiated particle energy and fluences. In polymer material, the g value corresponds to two free radicals, one is pyrolytic carbon free radicals, the other is a hydroxy superoxide radicals. The irradiation experiment was under a vacuum environment, there was not enough oxygen reacting with free radicals, so it was impossible to generate such a large number of hydroxyl superoxide radicals. Thus, most of the free radicals should be a pyrolytic carbon free radical. If free radicals were induced abundantly in main chains of PEEK, they would react in different ways to induce oxidation, crosslinking or degradation.

### 3.5. FTIR Analysis

The Fourier infrared spectrum of pristine and PEEK irradiated by 170 keV proton with different fluences are shown in Figure 7. With the increasing of the irradiation fluences, the intensity of all the characteristic absorption peaks gradually weakened and even disappeared. The absorbance of the ketone group (C=O) stretching vibration peaked at 1651 cm^−1^, the aromatic skeletal vibration occurred at 1598, 1490 and 1413 cm^−1^, the ether bond (C–O–C) was asymmetric stretching at 1280 and 1187 cm^−1^, aromatic hydrogen in-plane deformation bands occurred at 1157 and 1103 cm^−1^, the diphenyl ketone band occurred at 927 cm^−1^ and the plane bending modes of the aromatic hydrogens occurred at 860 and 841 cm^−1^, which was consistent with the previous reports on this material [24,25]. Low-energy proton irradiation has a great effect on the molecular structure of PEEK materials. It has been shown that heavy ion irradiation can destroy benzene rings and generate a large number of aromatic fragments according to the references [26]. It can be inferred that the low-energy proton irradiation will destroy the carbon skeleton of the benzene ring in the molecular structure of PEEK, resulting in the decrease of C=O and benzene content in PEEK. By comparing the intensity of the characteristic absorption peaks of the materials before and after the irradiation sources, it can be seen that the intensity of characteristic absorption peaks of each group decreased slightly with the increasing of the irradiation fluences. The reason for this change may be the dipole moment of the material changed due to irradiation, which improved the infrared scattering of the material surface and weakened the absorption intensity. The spectra were obviously changed by 1 × 10^16^ p/cm^2^ irradiation fluences. The transmittance decreased dramatically, in addition to the two oxidation-related peaks at 1730 cm^−1^ and 1100 cm^−1^. This is also attributed to the scission of C-H bonds and the following oxidation, which in turn destroyed chemically symmetric structures. This structural change would make PEEK hard and brittle, which is a typical mechanical change induced in PEEK aged by irradiation.

### 3.6. DSC Analysis

The DSC spectra of the pristine and PEEK irradiated with different fluences are shown in Figure 8. Table 3 shows the melting and crystallization parameters of PEEK before and after irradiation, including initial melting temperature T_1_, ending melting temperature T_2_, melting enthalpy ΔH_m_, starting crystallization temperature T_3_, ending crystallization temperature T_4_, crystallization enthalpy ΔH_c_ and crystallinity X(%). From the figure we can see that the melting peak of the PEEK was about 338 °C, the melting temperature of irradiated PEEK moved towards the higher temperature slightly, and the ΔH_m_ of irradiated PEEK decreased with the increasing of the fluences. The crystallization peak of PEEK was around 298 °C, the crystallization temperature of irradiated PEEK increased first than decreased slightly with the increasing of the fluences, and the crystallization enthalpy ΔH_c_ decreased slightly. The crystallinity of the polymer after irradiation reduced from 17% to 13%. The reason is that the crystal structure of PEEK changed after irradiation and the crystallinity decreased, causing the enthalpy of crystallization and melting enthalpy to decrease.

### 3.7. Stress–Strain Curves Analysis

The tensile stress–strain curves of the pristine and 170 keV proton irradiated PEEK with different fluences at room temperature and the inner figure, which is a locally enlarged view of yield strength, are shown in Figure 9. The tensile curves are divided into three stages including elastic deformation, cold drawing and strain hardening. The first elastic deformation stage was mainly the deformation of the molecular chain in the amorphous region, and the stress increased with the increasing of the strain until it reached the yield point. In the cold drawing stage, the necking cross-sectional area remains largely unchanged, but the necking part increased until the entire sample became thinner. As a result, the stress level at this stage was almost unchanged, and the strain increased gradually. In the strain hardening stage, the stress increased with the increasing of the strain, and the sample was uniformly stretched to the break point. It can be seen from the figure that the basic characteristics of the tensile stress-strain curves were almost the same between the pristine and the irradiated PEEK with 1 × 10^15^ p/cm^2^. The yield strength of the PEEK irradiated with 5 × 10^15^ p/cm^2^ and 1 × 10^16^ p/cm^2^ was higher than that of the pristine, but the elongation at break of the PEEK irradiated with 5 × 10^15^ p/cm^2^ and 1 × 10^16^ p/cm^2^ decreased obviously. The defects of the PEEK films after low energy proton irradiation increased and the molecular weight decreased. It can be seen from the results that, although the irradiation damage was only about 3 μm on the surface of the material, it had a great influence on the elongation at break of the entire material.

### 3.8. Discussion

The schematic of the irradiation process and irradiation degradation process and the EPR results of PEEK repeat unit is shown in Figure 10. The schematic shows that the PEEK films irradiated by the 170 keV proton may produce defects and the free radicals on the surface of the irradiated PEEK films. The accelerated 170 keV proton had enough energy to break all the chemical bonds in organic materials. The irradiation sources may break the benzene rings, ether bonds and C=O bonds of the PEEK films [27]. The most common result of the chemical bonds breakage is the formation of free radicals. From the EPR results, we can know that most of the free radicals were pyrolytic carbon free radicals. The free radicals may have combined with active factors such as oxygen when the irradiated materials were put in the air, which may have led to formation of defects and oxide layer on the surface of PEEK. The irradiation processes can be classified according to the effect of formation of free radicals, including curing, crosslinking, degradation and grafting. It can be seen from the stress-strain curves that the elongation at break of the material decreased after irradiation, so the most likely reason is that the molecular weight of the polymer decreased, and the irradiation caused the PEEK material to degrade. Because the damage was only on the surface of the polymer, the yield strength did not change significantly for the overall material. According to the GISAXS and FTIR results, the radiation had a significant effect on the surface structure damage. The DSC results show that irradiation will cause the crystallinity of PEEK to decrease.

## 4. Conclusions

PEEK films were irradiated by 170 keV protons and the microstructure and the mechanical properties of PEEK before and after proton irradiation were characterized. The SIRM calculation results showed that the irradiation caused vacancies in the material and the damage thickness was about 3 μm. The FTIR results showed that the intensity of all the characteristic absorption peaks gradually weakened and even disappeared with the increasing of the irradiation fluences. The EPR results showed that PEEK produced a lot of free radicals after irradiation, and the content of free radicals increased with the increase of irradiation fluences. The DSC results showed that the crystallinity of the polymer after irradiation decreased. The results of GISXAS showed that the pristine PEEK samples had peaks and the peaks changed with the increasing of angle of incidence. After the irradiation of the 170 keV protons, the scattering patterns and peaks changed gradually. The yield strength of the irradiated PEEK with 5 × 10^15^ p/cm^2^ and 1 × 10^16^ p/cm^2^ increased a little, which compared to that of the pristine, but the elongation at break of the irradiated PEEK with 5 × 10^15^ p/cm^2^ and 1 × 10^16^ p/cm^2^ decreased obviously.

## Figures and Tables

**Figure 1 polymers-12-02717-f001:**
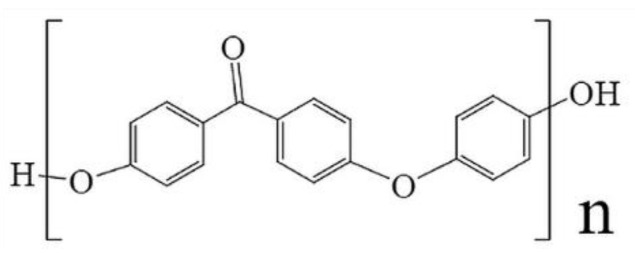
The Schematic of the repeat unit of polyether ether ketone (PEEK).

**Figure 2 polymers-12-02717-f002:**
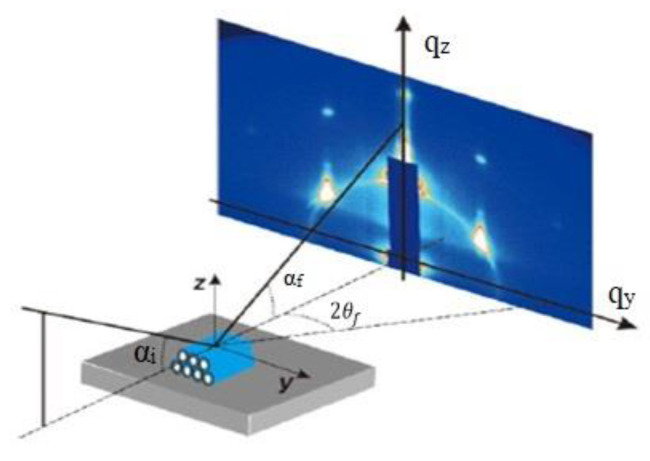
Schematic diagram of GISAXS test.

**Figure 3 polymers-12-02717-f003:**
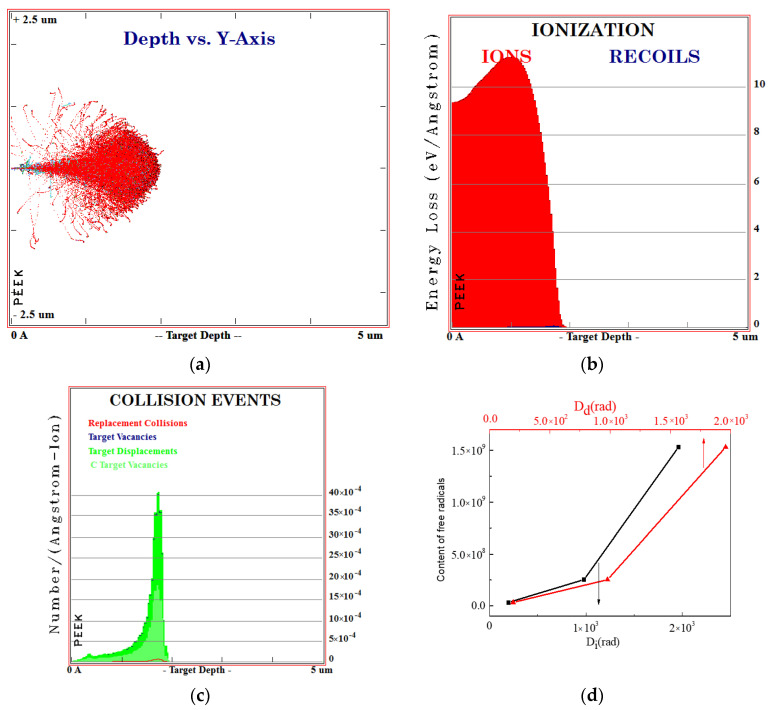
SRIM simulation of PEEK irradiated with 170 keV protons. (**a**) Simulation of particle collision process; (**b**) simulation of ionization energy loss of ion collision process; (**c**) simulation of vacancy distribution after ion irradiation; (**d**) content of free radicals change with ionization and displacement absorbed dose.

**Figure 4 polymers-12-02717-f004:**
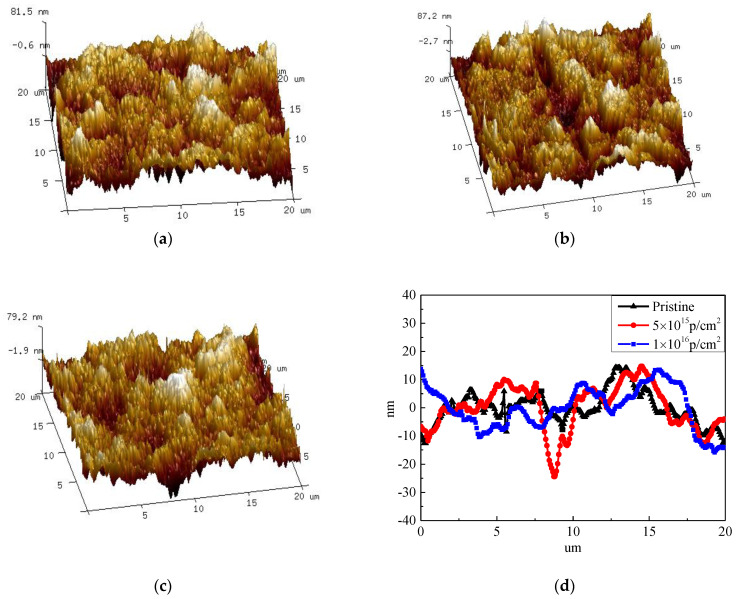
Surface roughness morphology of (**a**) pristine PEEK, (**b**) PEEK irradiated with 5 × 10^15^ p/cm^2^, and (**c**) PEEK irradiated with 1 × 10^16^ p/cm^2^, and (**d**) the surface roughness of the pristine and irradiated PEEK.

**Figure 5 polymers-12-02717-f005:**
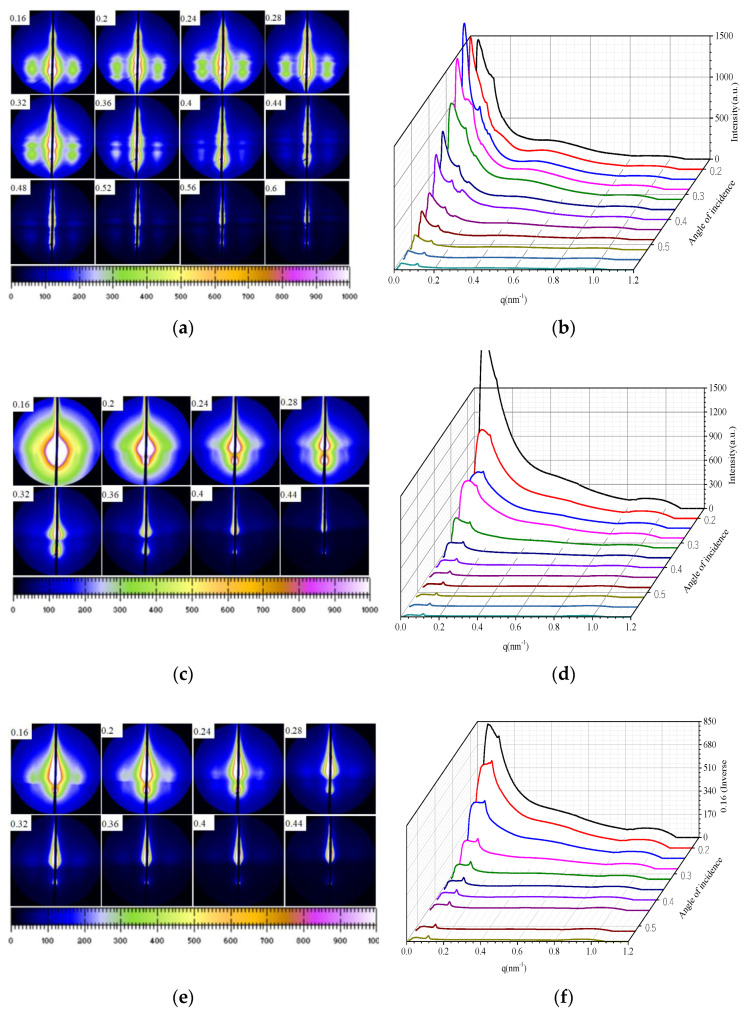
The different GISAXS scattering patterns and intensity distribution curves of pristine and irradiated PEEK with the incident angle, (**a**,**b**) pristine, (**c**,**d**) 5 × 10^15^ p/cm^2^, (**e**,**f**) 1 × 10^16^ p/cm^2^.

**Figure 6 polymers-12-02717-f006:**
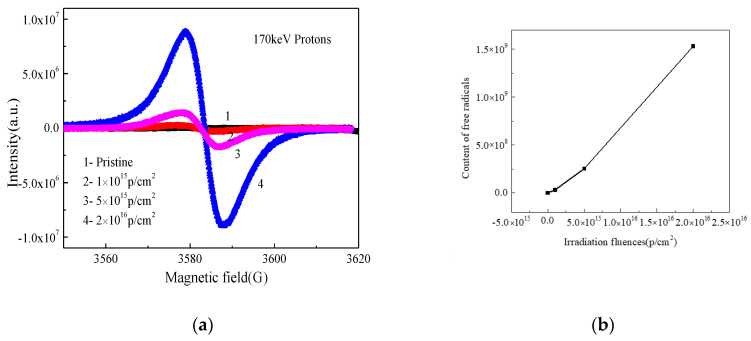
(**a**) The EPR spectrogram of pristine and irradiated PEEK with different fluences, (**b**) the content of free radicals changed with fluences.

**Figure 7 polymers-12-02717-f007:**
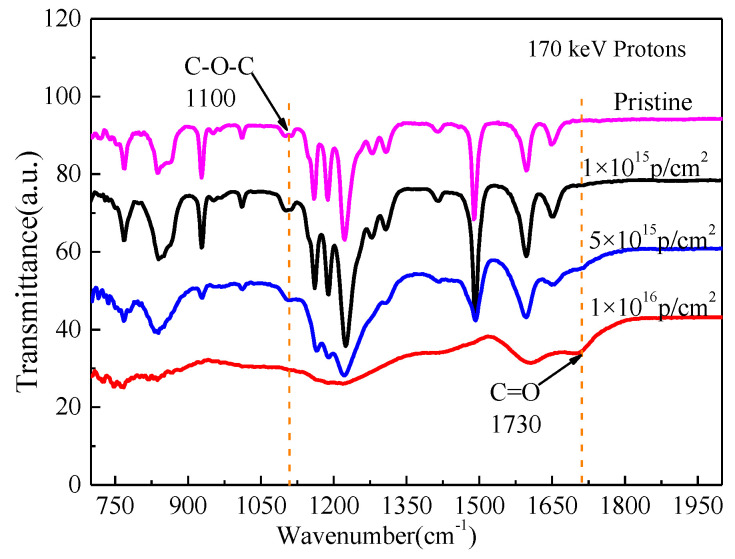
FT-IR spectra of pristine and PEEK irradiated with different fluences.

**Figure 8 polymers-12-02717-f008:**
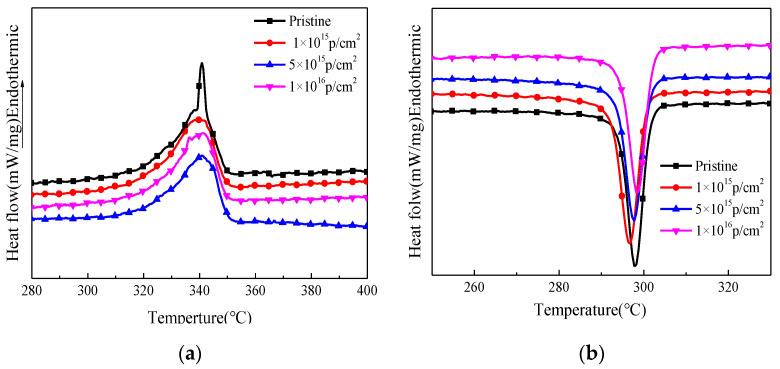
The DSC spectra of the pristine and PEEK irradiated with different fluences, (**a**) The melting temperature; (**b**) The crystallization temperature

**Figure 9 polymers-12-02717-f009:**
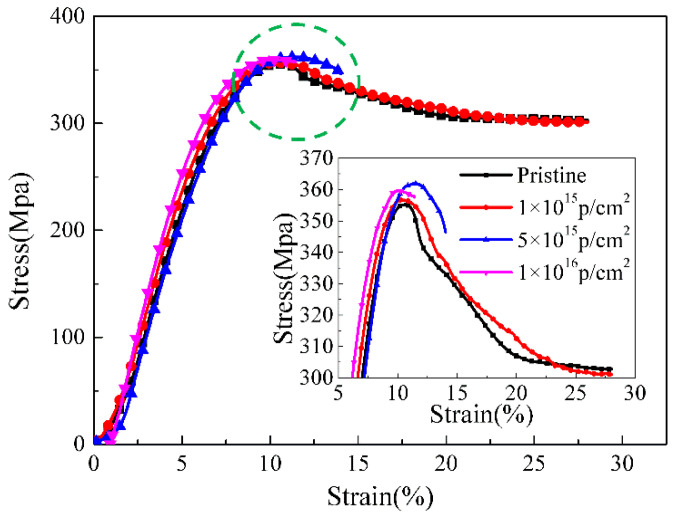
Stress–strain curves of pristine and PEEK films irradiated with different fluences and the inner image which is the local enlarged image.

**Figure 10 polymers-12-02717-f010:**
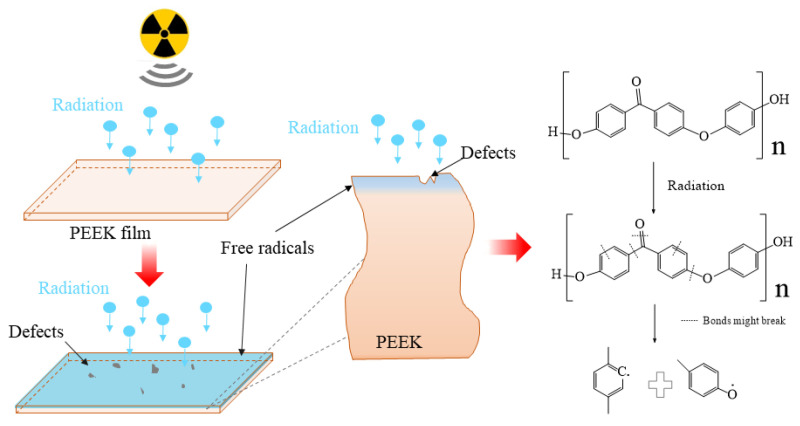
Schematic of the irradiation process and irradiation degradation process and the EPR results of PEEK repeat unit.

**Table 1 polymers-12-02717-t001:** Values of roughness parameters (Ra and Rq) of the PEEK films irradiated with different fluences.

Sample	Pristine	5 × 10^15^ p/cm^2^	1 × 10^16^ p/cm^2^
Ra(nm)	18.8	20.8	17.5
Rq(nm)	23.4	26.1	22.3

**Table 2 polymers-12-02717-t002:** The changes in long period of the pristine and the irradiated PEEK with the incident angle.

Sample	0.16	0.2	0.24	0.28	0.32	0.36	0.4	0.44	0.48
L(nm)/Pristine	16.44	16.10	16.02	15.90	15.66	15.43	15.24	14.74	14.6
L(nm)/5 × 10^15^ p/cm^2^	14.78	14.6	14.37	13.86	13.14	12.41			
L(nm)/1 × 10^16^ p/cm^2^	14.4	14.08	13.96	13.36	12.56				

**Table 3 polymers-12-02717-t003:** Melting and crystallization parameters of PEEK before and after irradiation.

Sample	Pristine	5 × 10^15^ e/cm^2^	1 × 10^16^ e/cm^2^
T1 (°C)	301.8 ± 0.5	305.5 ± 0.5	309.2 ± 0.5
T2 (°C)	350.9 ± 0.5	351.3 ± 0.5	352.7 ± 0.5
ΔH_m_ (J/g)	21.9 ± 5	20.4 ± 5	16.7 ± 5
T3 (°C)	301.4 ± 0.5	302.5 ± 0.5	303.4 ± 0.5
T4 (°C)	294.5 ± 0.5	296.7 ± 0.5	299.9 ± 0.5
ΔH_c_ (J/g)	40.9 ± 5	43.0 ± 5	39.5 ± 5
X (%)	17	15.8	13

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
