# Peer review of "Study on the Microstructure of Polyether Ether Ketone Films Irradiated with 170 keV Protons by Grazing Incidence Small Angle X-ray Scattering (GISAXS) Technology"

_polymers, 2020, doi:10.3390/polym12112717_

Round 1
Reviewer 1 Report
This research is very interesting because the authors studied the PEEK films irradiated with 170 keV protons and concluded the semi-crystalline PEEK were irradiated by 170 keV protons, the micro-structure and the 324 mechanical properties of PEEK before and after proton irradiation were characterized.
The reviewer has minor comments.
- The resolution of figure 2 needs to improve.
- English improvement is necessary for the manuscript.
- How do the authors correlate the PEEK films and semi-crystalline PEEK?
Author Response
Dear Editor,
Thank you very much for your reply and help. Thanks a lot for the reviewers’ comments and their kind suggestions of our manuscript (Polymers-1003280) entitled “Study on the Micro-structure of Polyether ether ketone Films irradiated with 170 keV Protons by GISAXS technology”. We have revised the article as the academic editor and reviewers’ suggestions. In order to make the changes easily viewable for you and reviewers, in the revised manuscript, we marked the revisions with red color and used “Track Changes”function in the MS as you suggest. We have checked carefully every sentences of the whole manuscript and corrected the grammar mistakes. We hope the revised paper would satisfy you and the reviewers.
We are looking forward to hearing from you soon.
Kind Regards,
Xingji li
..........................................................................................................................................
Revision list according to the comments
Reviewer: 1. This research is very interesting because the authors studied the PEEK films irradiated with 170 keV protons and concluded the semi-crystalline PEEK were irradiated by 170 keV protons, the micro-structure and the mechanical properties of PEEK before and after proton irradiation were characterized.
The reviewer has minor comments.
- The resolution of figure 2 needs to improve.
Reply:
Thank you for your remind. We have revised the Figure 2 to be clear. We copy here for your check.
- English improvement is necessary for the manuscript.
Reply:
Thank you for your suggest. We have checked carefully every sentences of the whole manuscript and corrected the grammar mistakes. We copy some here for your check.
Line 10-13: The PEEK films irradiated with 170 keV protons were calculated by the stopping and ranges of ions in matter (SRIM) software. The results showed that the damage caused by 170 keV protons was only several microns of the PEEK surface, and the ionization absorbed dose and displacement absorbed dose were calculated.
Line 22-25: The stress and strain test results showed that the yield strength of the PEEK irradiated with 5×1015 p/cm2 and 1×1016 p/cm2 was higher than the pristine, but the elongation at break of the PEEK irradiated with 5×1015 p/cm2 and 1×1016 p/cm2 decreased obviously.
Line73-74: In this paper, the surface micro-structure damage mechanisms of the PEEK irradiated with 170 keV proton were studied.
- How do the authors correlate the PEEK films and semi-crystalline PEEK?
Reply:
Thank you for your comment. The PEEK films is a kind of semi-crystalline polymer. Thus the PEEK films is the semi-crystalline PEEK. To avoid misunderstanding, we changed the semi-crystalline PEEK films to PEEK films, as shown Line 330. We copy here for your check.
The PEEK films were irradiated by 170 keV protons, the micro-structure and the mechanical properties of PEEK before and after proton irradiation were characterized.

Reviewer 2 Report
-Minor revision in English is required
-Figure 4 denotes that AFM revealed no differences (considering statistical errors and different selected areas)
-same for FTIR; authors acknowledge this, thus it is suggested that since the extent of affected zone is not quantified, a technique such as nanoindentation could reveal this extent.
Author Response
Dear Editor,
Thank you very much for your reply and help. Thanks a lot for the reviewers’ comments and their kind suggestions of our manuscript (Polymers-1003280) entitled “Study on the Micro-structure of Polyether ether ketone Films irradiated with 170 keV Protons by GISAXS technology”. We have revised the article as the academic editor and reviewers’ suggestions. In order to make the changes easily viewable for you and reviewers, in the revised manuscript, we marked the revisions with red color and used “Track Changes”function in the MS as you suggest. We have checked carefully every sentences of the whole manuscript and corrected the grammar mistakes. We hope the revised paper would satisfy you and the reviewers.
We are looking forward to hearing from you soon.
Kind Regards,
Xingji li
..........................................................................................................................................
Reviewer: 2. 1. Minor revision in English is required
Reply:
Thank you for your suggest. We have checked carefully every sentences of the whole manuscript and corrected the grammar mistakes. We copy some here for your check.
Line 10-13: The PEEK films irradiated with 170 keV protons were calculated by the stopping and ranges of ions in matter (SRIM) software. The results showed that the damage caused by 170 keV protons was only several microns of the PEEK surface, and the ionization absorbed dose and displacement absorbed dose were calculated.
Line 18-20: The influences of PEEK irradiated with protons on the melting temperature and crystallization temperature was investigated by differential scanning calorimetry (DSC). The DSC results showed that the crystallinity of the polymer after irradiation decreased.
Line 127-129: Understanding the sensitivity of polymer materials to ionization/displacement effect is of great significance for predicting the performance of polymer materials in a irradiation environment.
- -Figure 4 denotes that AFM revealed no differences (considering statistical errors and different selected areas)
Reply:
Thank you for your suggest. The AFM results showed that the surface roughness of the PEEK before and after irradiation changed slight. Actually, roughness has a complicated effect on the results of GISAXS, the AFM results for ignoring the influence of different roughness on the GISAXS results. We revised the description, as shown Line 176-178. We copy here for your check.
But overall, as can be seen from Table 1, the surface roughness of PEEK before and after irradiation change slight. Generally speaking, the roughness of the material affects the GISAXS results, this GISAXS results ignore the effect of the roughness.
- -same for FTIR; authors acknowledge this, thus it is suggested that since the extent of affected zone is not quantified, a technique such as nanoindentation could reveal this extent.
Reply:
Thank you for your suggest. The nanoindentation technology is effective method to analyze the surface of the material. But the equipment is under maintenance in our school unfortunately. We have no enough time to test in other institute due to only 5 days need submit the revised manuscript. We will use it in the future research and thank you for your understanding. In addition, we calculated the incident thickness of the material using SRIM. The advanced GISAXS technology tested the changes of the material surface structure before and after irradiation at different incident angles. This is a good explanation of the changes in the surface structure of the material.

Reviewer 3 Report
Dear Editor: I would like to express my deep thanks for inviting me to review the manuscript ID: polymers-1003280
Title: Study on the Micro-structure of Polyether ether ketone Films irradiated with 170 keV Protons by GISAXS technology
Authors: Hongxia Li, Jianqun Yang, Feng Tian, Xingji Li and Shangli Dong
Comments:
Abstract:
Please rewrite the abstract according to your results for example mention the value of stress and strain test results.
Introduction part:
Please write the aim and novelty in this work at the end of introduction section.
Materials and methods:
Please check it “with a m2ultimode scanning probe”.
Please include the number tensile test specimens
Results and discussion:
- The quality of Figure 4 is very poor. Please add better quality images.
- Mention the number of samples tested for roughness measurement.
- Please check all peaks in FTIR data in Figure 7.
- Please provide clear image in 8(b) specially in peak region.
RECOMMENDATION
After reviewing the enclosed manuscript for “Polymers”, the present manuscript contains some kinds of scientific analysis but it is mandatory required to modify according to the preceding remarks. So, the manuscript can be accepted for publication after major mandatory revisions have been made.
Author Response
Dear Editor,
Thank you very much for your reply and help. Thanks a lot for the reviewers’ comments and their kind suggestions of our manuscript (Polymers-1003280) entitled “Study on the Micro-structure of Polyether ether ketone Films irradiated with 170 keV Protons by GISAXS technology”. We have revised the article as the academic editor and reviewers’ suggestions. In order to make the changes easily viewable for you and reviewers, in the revised manuscript, we marked the revisions with red color and used “Track Changes”function in the MS as you suggest. We have checked carefully every sentences of the whole manuscript and corrected the grammar mistakes. We hope the revised paper would satisfy you and the reviewers.
We are looking forward to hearing from you soon.
Kind Regards,
Xingji li
..........................................................................................................................................
Reviewer: 3.
- Abstract:
Please rewrite the abstract according to your results for example mention the value of stress and strain test results.
Reply:
Thank you for your suggest. We have revised the abstract, as shown Lin 22-25. We copy here for your check.
The stress and strain test results showed that the yield strength of the PEEK irradiated with 5×1015 p/cm2 and 1×1016 p/cm2 was higher than the pristine, but the elongation at break of the PEEK irradiated with 5×1015 p/cm2 and 1×1016 p/cm2 decreased obviously.
- Introduction part:
Please write the aim and novelty in this work at the end of introduction section.
Reply:
Thank you for your suggest. We have revised the manuscript as your opinion, as shown Line 53-58, Line 73-80. We copy here for your check.
There are a large number of low-energy changed particles in space, and these particles cause great damage to the structure and performance of the material. The damage caused by low-energy proton irradiation is only on the surface. How the surface structure and properties change after irradiation and the impact on the overall structure and properties have not been studied yet. The changes in the internal crystals of the material will significantly affect the changes in material properties. The effect of surface structure damage on the overall structure and performance degradation of the material is of great significance to the study of material degradation mechanisms.
In this paper, the surface micro-structure damage mechanisms of the PEEK irradiated with 170 keV proton were studied. The tensile tests and AFM were used to analyze the stress-strain and surface morphology of PEEK. Synchrotron radiation grazing incidence small angle X ray scattering (GISAXS), FT-IR and DSC were applied to analyze the structure and crystallinity change of the PEEK after 170 keV proton irradiation. We used SRIM to calculate the incident depth and ionization and displacement absorbed dose of the PEEK, combined the advanced GISAXS technology and other experimental methods to reveal the influence of the material surface damage on the overall structure and performance degradation.
- Materials and methods:
Please check it “with a m2ultimode scanning probe”.
Please include the number tensile test specimens
Reply:
Thank you for your suggest. We have revised the mistake, as shown Line 93. We add the number test specimens, as shown Line 91. We copy here for your check.
The AFM images were obtained by atomic force microscope with a multimode scanning probe microscope (Dimension Fastscan) produced from Bruker company, Germany.
The size for the four dumbbell-shaped tensile samples are 12, 4 and 0.5 mm, respectively.
- Results and discussion:
The quality of Figure 4 is very poor. Please add better quality images.
Reply:
Thank you for your suggest. The AFM figure exported from NanoScope Analysis software, we can just change the dpi of the figure, but the quality images is same with that one. This quality images is clear enough to see the surface roughness morphology.
Mention the number of samples tested for roughness measurement.
Reply:
Thank you for your suggest. The number of the samples is showed in the results, including (a) pristine PEEK, (b) PEEK irradiated with 5×1015p/cm2, (c) PEEK irradiated with 1×1016p/cm2, the three samples were tested, every samples tested many place and choose good results to show here.
Please check all peaks in FTIR data in Figure 7.
Reply:
Thank you for your suggest. The figure will be complicated and messy if we check all peaks in the Figure 7.
Please provide clear image in 8(b) specially in peak region.
Reply:
Thank you for your good suggest. We have revised the Fig.8(b). We copy here for your check.
- RECOMMENDATION
After reviewing the enclosed manuscript for “Polymers”, the present manuscript contains some kinds of scientific analysis but it is mandatory required to modify according to the preceding remarks. So, the manuscript can be accepted for publication after major mandatory revisions have been made.
Reply:
Thank you for your suggest. We have revised the manuscript carefully and check the grammar mistakes. We hope it can satisfy you.

Round 2
Reviewer 3 Report
Authors addressed all comments in revised manuscript.